# Changes in Polychlorinated Biphenyl Residues in Milk during Lactation: Levels of Contamination, Influencing Factors, and Infant Risk Assessment

**DOI:** 10.3390/ijms232112717

**Published:** 2022-10-22

**Authors:** Agata Witczak, Anna Pohoryło, Aleksandra Aftyka, Kamila Pokorska-Niewiada, Grzegorz Witczak

**Affiliations:** 1Department of Toxicology, Dairy Technology, and Food Storage, West Pomeranian University of Technology in Szczecin, Papieża Pawła VI Street 3, 71-459 Szczecin, Poland; 2Veterinary Inspection Provincial Veterinary Inspectorate in Szczecin, 71-337 Szczecin, Poland; 3Department of Gynecology, Endocrinology and Gynecological Oncology, Pomeranian Medical University in Szczecin, 71-252 Szczecin, Poland

**Keywords:** polychlorinated biphenyls, human milk, infant health, stages of lactation

## Abstract

Given the importance of breastfeeding infants, the contamination of human milk is a significant public concern. The aim of this study was to assess the contamination of human milk with dioxin-like PCBs (dl-PCBs) and non-dioxin-like PCBs (ndl-PCBs) in relation to the duration of lactation and other influencing factors, especially the frequency of the consumption of selected foods during pregnancy. Based on this, the health risk to infants was assessed and compared to the tolerable daily intake (TDI). PCB determinations were performed using gas chromatography/mass spectrometry. The ∑ndl-PCB content ranged from 0.008 to 0.897 ng/g w.w., at an average of 0.552 ng/g wet weight, which was 55% of the maximum level according to the EU guidelines for foods for infants and young children. The toxic equivalent (TEQ) was in the range of 0.033–5.67 pg-TEQ/g w.w. The content of non-ortho, mono-ortho, and ndl-PCBs in human milk decreased the longer lactation continued. Moreover, when pregnant women smoked tobacco, this correlated significantly with increases in the concentrations of PCB congeners 156, 118, and 189 in human milk. The human milk contents of PCB congeners 77, 81, 186, 118, and 189 were strongly positively correlated with the amount of fish consumed. The content of stable congeners PCB 135 and PCB 153 increased with age.

## 1. Introduction

According to the Stockholm Convention on Persistent Organic Pollutants (POPs), polychlorinated biphenyls (PCBs) are among the most important pollutants. The World Health Organization (WHO) introduced worldwide measurement campaigns to determine infant exposure to dioxin-like polychlorinated biphenyls (dl-PCBs) [1]. These compounds are hydrophobic and lipophilic, and their water solubility decreases as the degree of molecular chlorination increases. PCBs can still be found at all links in trophic chains as a result of their widespread use in industry in the twentieth century, their atmospheric transport, and their high persistence [2].

Since these compounds are not readily biodegradable, long-term environmental exposure to them can contribute to health problems including damage to the liver, spleen, and kidneys, and they can lead to cancer [3]. PCBs are also known to be endocrine disruptors [4,5]. Triiodothyronine and thyroxine molecules have similar structures to certain PCBs, leading to potential thyroid hormone disruptions. PCB congeners 126 and 153 have also been shown to negatively affect the production of sex hormones by the ovaries. Additionally, PCB 126 increases testosterone levels, which results in the formation of ovarian cysts. These compounds also exhibit an ability to disrupt the proper functioning of the pituitary gland and the hypothalamus. Biphenyl molecules are structurally similar to estradiol, which means they can bind with its receptors in cytoplasm and penetrate cell nuclei [5]. In women, PCBs might be a cause of endometriosis, while in men they can lead to a decrease in the number and quality of sperm that worsens over time [6,7].

Underscoring the estrogenic, anti-estrogenic, and anti-androgenic effects of PCBs, many researchers suggest that PCBs could affect the development of the characteristics that determine the sex of children [8,9].

Infants first come into contact with organochlorine compounds (OCs), including PCBs, in the prenatal period, when these compounds cross the placental barrier.

The amount of penetration through the placental barrier is influenced by the concentration of xenobiotics in the mother’s blood serum, the surface of the placenta, and the thickness of the blood–placenta barrier [10,11]. The primary source of these compounds in infants, however, is their mothers’ milk, which, in addition to essential nutritional components, also contains undesirable contaminants, especially those that are readily fat-soluble, for example PCBs [12,13].

Data from the literature indicate that even though many years have passed since products containing PCBs have been produced in most countries, the contents of dl-PCBs in human milk have decreased exceptionally slowly [14,15,16]. Risk assessment indicates that the levels of dibenzo-p-dioxins (PCDDs), polychlorinated dibenzofurans (PCDFs), and PCBs in human milk remain significantly higher than those believed to be toxicologically safe [3]. It is estimated that PCB residues will be found above the limits of detection in living organisms for subsequent generations. Because of their physiologically lower enzyme activity, infants and small children are significantly more susceptible to the toxic effects of OCs [17,18].

Human milk is a useful matrix for PCB biomonitoring: although it omits a large portion of the population, it is readily available, collecting it is non-invasive, and its high lipid content facilitates the extraction of POPs [19].

The various factors that can affect the accumulation of PCBs in breast milk include the following: place of residence; smoking; age and body weight of the mother; and the duration of any previous lactation [20]. The diets of pregnant women might also be important.

Even trace concentrations of dl-PCBs and non-dioxin-like PCBs (ndl-PCBs), such as those found in foods and in the diets of prenatal and postpartum women, can cause subtle effects, especially when exposure is long-term [21,22].

Currently, there are no explicit legal regulations specifying the maximum permissible concentrations of PCB residues in human milk.

The present study attempted to estimate changes in non-ortho, mono-ortho, and ndl-PCB congeners in human milk at different stages of lactation. The influence of other factors, such as the mother’s age and weight, place of residence, parity, and nutritional habits during pregnancy, on the PCB levels in milk was also analyzed.

This assessment of infant exposure to residues of ndl-PCBs and dl-PCBs in human milk is the first study of its kind in northwestern Poland. In this study, we also assumed that the concentration of selected polychlorinated biphenyls in human breast milk may have an association with the individual features of the mother and her infant, and especially the duration of lactation and the frequency of the consumption of selected foods during pregnancy.

## 2. Results

### 2.1. Characteristics of Study Participants and Their Diets during Pregnancy

The analysis of participant age indicated that the majority (37.5% of those participating) were 28–32 years old, while 33.3% of the participants were 33–36 years old. Most participants (83.3%) resided in cities with populations exceeding 5000, while 12.5% of the participants resided in smaller towns. Most of the women (95.8%) had higher educations (Table 1).

Maternal body weight both before and after giving birth varied from less than 55 kg to more than 65 kg. The number of natural births among the participants was the same as the number of cesarian sections, while 62.5% of the participants were primiparous, and 37.5% were multiparous (parity ranged from two to a maximum of three) (Table 1).

Based on the data from the questionnaire, 62.5% of the new-born infants were male, while 37.5% were female. The body weight of the new-born infants in their first week of life ranged from 2.4 kg to over 4 kg (Table 1). The weight range among male new-born infants was greater at 2.4–4 kg, while the female new-born infant weight range was 2.9–3.6 kg.

The smoking of tobacco among the pregnant women was also analyzed. Most of the women (75%) declared on the questionnaire that they had never smoked cigarettes, while 16.7% quit smoking before becoming pregnant. Disturbingly, as many as 4.2% of the women participating in the study smoked one or more packs of cigarettes daily (Table 1). The questionnaire on nutritional habits indicated that almost all the women consumed fish during pregnancy (95.9%) (Figure 1).

High dairy product consumption was also noted. The questionnaire also asked about the consumption of various types of meat, and the responses of the participants indicated that there was a high consumption of beef, pork, and poultry. Over 70% of the pregnant women declared that they consumed poultry more than twice weekly. Approximately 80% of the women declared that they consumed vegetables daily, while 70% of them declared that they consumed fruit daily. Half of the women declared that they consumed cooked whole grains less often than once weekly, and only 4.2% of the women declared that they consumed these types of products more frequently than twice weekly (Figure 1).

### 2.2. Composition of Human Milk during Lactation

Decreases were observed in the dry matter, lipids, and protein content of human milk throughout lactation with negative r correlations (*p* < 0.05) of −0.42, −0.38, and −0.50, respectively (Figure 2).

The dry matter content ranged from 8.66 ± 0.3% to 15.34 ± 0.1%. Taking into account the interrelationships between milk components, a weak positive correlation was observed between the dry weight of milk and the weight of the child in the first week after delivery (r = 0.24). Average positive relationships between dry matter and protein content were also found, where r = 0.43, and a positive high correlation between the concentration of fat in milk and the content of dry matter (r = 0.69). Additionally, a weak positive correlation between dry weight and lactose concentration was noted (r = 0.23).

The average fat content in all analyzed lactation periods was 3.83 ± 0.52%; the lowest concentration was observed in milk obtained after more than one year of lactation (2.37 ± 0.17%), and the highest concentration in milk collected 7 days postpartum (4.60 ± 0.77%). We also observed weak positive correlations between the content of fat in milk and the number of deliveries experienced (r = 0.16) and between the concentration of fat in milk and the birth weight of the infant in the first week of life (r = 0.20). In addition, the fat content on average positively correlated with the content of protein and lactose: respectively, r = 0.47 and r = 0.32. On the other hand, there was no effect of the mother’s age and body weight after pregnancy on the fat concentration in the tested milk.

The protein content ranged from 0.84 ± 0.06% to 1.49 ± 0.23%. There was no effect of the number of births and the mother’s body weight, both before and after pregnancy, on the protein content in milk.

The lowest concentration of lactose was recorded in the milk collected in the first week of lactation (5.33 ± 0.30%), while the highest was recorded after 7 days of feeding the infant (7.27 ± 0.47%). It was also noticed that the concentration of lactose was slightly positively correlated with the protein content (r = 0.19) and the dry matter content (r = 0.28). However, there was no correlation between the level of lactose in the milk and the age of the mother, the number of births, the mother’s body weight before and after pregnancy, or the infant’s birth weight.

Weak negative correlations were noted between the stages at which milk was collected and its lactose content (r = −0.15) (Figure 2).

### 2.3. Changes in PCB Contents in Human Milk throughout the Duration of Lactation

#### 2.3.1. ndl-PCBs

The content of total indicator congeners (ndl-PCBs) in human milk fluctuated from 0.343 ± 0.035 ng/g lipids to 20.31 ± 1.36 ng/g lipids. Decreases in the content of ndl-PCBs were noted throughout lactation, while the highest concentrations were in the first month of breastfeeding (Table 2).

Weak average negative correlations (*p* < 0.05) were noted between the content of individual ndl-PCB congeners and the duration of lactation, with correlation coefficient r values ranging from −0.448 for PCB 52 to −0.280 for PCB 153. No such relationship was noted for PCB 101 (Figure 3).

As milk lipid levels increased, so did the concentrations of most ndl-PCB congeners: r _PCB 28_ = 0.328, r _PCB 52_ = 0.319, r _PCB 101_ = 0.101, r _PCB 153_ = 0.658, r _PCB 138_ = 0.602, and r _PCB 180_ = 0.428.

#### 2.3.2. dl-PCBs

non-ortho PCBs

Total non-ortho PCB congeners in milk throughout lactation ranged from 0.212 ± 0.037 to 14.85 ± 1.19 ng/g lipids. The lowest concentration of total non-ortho PCBs was noted in milk collected after 12 months of lactation, while the highest were noted 7 and 14 days after childbirth (Table 2). PCB 169 dominated among the non-ortho congeners. The content of non-ortho PCBs in milk decreased throughout lactation, and the majority of the pool of these compounds accumulated in the maternal body were excreted into the milk during the initial breastfeeding period (Figure 4a).

A weak average positive correlation (r = 0.202–0.481) was confirmed between milk lipid contents and non-ortho PCB residues, which proved not only the lipophilic nature of these compounds, but also the influence of other factors that can affect their presence in human milk (Figure 4b).

mono-ortho PCBs

The mean content of total mono-ortho PCBs was 5.88 ± 4.09 ng/g lipids (Table 2). The highest concentration of these compounds was recorded after 7 days of lactation, while the maximum concentration of PCB 189 was noted after 14 days of lactation. PCBs 105, 114, 167, 156, 118, 157, and 189 were not detected in either the wet matter or the lipids of milk collected after more than one year of lactation. The concentrations of mono-ortho PCB congeners decreased throughout lactation, although this trend was barely notable in some instances (r = −0.533–0.190) (Figure 5).

Positive relationships were also noted between the content of mono-ortho PCB congeners and milk lipid content (r = 0.20–0.59) (Figure 6).

### 2.4. Redundancy Analysis (RDA)—Influence of the Frequency of Pregnant Women’s Consumption of Foodstuffs and Changes in PCB Contents in Human Milk

Redundancy analysis indicated a directly proportional relationship between the content of PCB congeners 180, 101, 52, and 28 in milk and the quantities of dairy products consumed and maternal weight before pregnancy. The concentrations of these congeners negatively correlated with the frequency of poultry, egg, and fish consumption. Contrary to the study assumptions, the concentrations of PCBs 153 and 138 in milk positively correlated with parity and the content of lipids and dry matter in the milk, but they negatively correlated with women smoking cigarettes prior to pregnancy. This dependence may have also resulted from the influence of other factors, e.g., the duration of active or passive exposure to smoke or the time since quitting smoking. Negative correlations were also noted with the weight of women prior to pregnancy and the quantities of fish and eggs consumed (Figure 7).

The contents of both PCB 77 and PCB 81 in milk was strongly positively correlated with the quantities of fish consumed, while the concentrations of these compounds decreased with the frequency of pork and dairy product consumption. RDA analysis also showed a positive correlation between the content of PCB 126 residues and the protein content of milk. The frequency of the consumption of wheat-rye bread negatively correlated with PCB 126 content (Figure 8).

Similarly to PCB 81, the content of the congener PCB 169 was largely influenced by all the factors analyzed. A strong inversely proportional dependence of PCB 169 on the duration of lactation was also noted (Figure 8).

RDA analysis (Figure 9) confirmed positive correlations between the concentrations of PCB congeners 123 and 167 and women’s residence in cities with populations greater than 5000.

It was also noted that the concentrations of PCB congeners 156, 118, and 189 in milk positively correlated with infant birth weight, women smoking prior to pregnancy, and the quantities of fish consumed. However, the content of these compounds decreased with the consumption of dairy products. The concentrations of PCB 114 and PCB 105 positively correlated with milk lipid content, dry matter, and lactose. Negative correlations were also noted between the content of mono-ortho PCB congeners in milk and the quantity of beef consumption (Figure 9).

### 2.5. Infant Risk Assessment Linked with the Content of PCBs in Human Milk

Since there are no other guidelines for the concentrations of PCB congeners, the results of the current study were compared to the maximum levels permitted in foods for infants and young children. Commission regulation (EU) No 1259/2011 [24] sets the maximum levels at 0.1 pg-TEQ/g w.w. (wet weight) for dl-PCBs and 1 ng/g w.w. for ndl-PCBs. The fluctuations in total ndl-PCBs throughout lactation (0.008–0.897 ng/g w.w.) noted in the current study results indicated that the maximum levels permitted were exceeded.

The dl-PCB content expressed as the toxic equivalent (TEQ) was within the range of 0.033–5.676 pg-TEQ/g w.w. Assuming that “foods for infants and young children” [24] also includes human milk, the maximum levels permitted for dl-PCBs were exceeded. When the different lactation stages were considered, the maximum levels were not exceeded only in milk collected after one year of breastfeeding (Figure 10).

A risk assessment of the daily intake of dl-PCBs in human milk was performed considering the differences in the quantities of milk consumed depending on infant sex and weight gain. The assumption was that male infants consumed an average of 76 mL milk more than female infants. Daily infant milk consumption ranged from 478 mL to 1356 mL, at an average of 798 mL [25,26,27].

Considering these data, the weekly intake of dl-PCBs in human milk was 0.021–7.33 pg-TEQ/kg bw/day, with an average intake of 2.13 pg-TEQ/kg bw/week for female infants and 2.18 pg-TEQ/kg bw/week for male infants (Figure 11).

Based on research by the International Agency for Research on Cancer [28], the classification of PCB compounds was changed from group 2A “probably carcinogenic to humans” to group 1 “carcinogens”, and the TWI values were changed from 14 pg-TEQ/kg bw/week to 2 pg-TEQ/kg bw/week [29].

## 3. Discussion

Many factors can influence the content of PCBs in human milk. Some researchers suggest that these include maternal age, maternal body weight, and parity [30,31,32,33,34]. One of the most frequently mentioned factors that could influence the levels of PCBs in human milk is maternal age. However, no such significant correlation was noted in the current study, which Klinčić et al. [35] also reported.

Another factor influencing PCB concentrations in human milk is maternal weight prior to pregnancy. Hassine et al. [36] confirmed a negative correlation between maternal body weight and PCB 138 concentrations. In the current study, there was a weak negative correlation between the content of this congener and body weight. While similar conclusions were also reported by other researchers [37,38,39], the redundancy analysis of the results of the present study confirmed increased concentrations of PCBs 28, 52, 101, 180, 118, 156, and 189 in milk with increasing maternal weight prior to pregnancy.

The current study results confirmed that maternal nutritional habits also significantly influenced PCB concentrations in human milk. According to Mamontova et al. [40], the increased exposure of women to maternal PCBs could result from the increased consumption of local foodstuffs. PCB levels in the milk of women who consumed foods obtained directly from farms or purchased at local markets were two to four times higher in comparison to those of women who purchased foodstuffs in supermarkets [40]. The respondents of this study lived in the region of northwestern Poland, which is located on the Baltic Sea. There are many fish processing plants in this area. It can therefore be argued that the consumed fish is a local product, the consumption of which may result in the increased content of PCBs in milk.

The main source of the exposure of the general population (apart from occupational exposure) to PCBs is food (90–95%). The presence of these compounds in food, including human milk, results from the infiltration of these pollutants from the environment. The share of dominant sectors in PCB emissions in 2020 in Poland was as follows: fuel combustion 76.94% (energy industry 69.2%, manufacturing industry and construction 5.6%, transport 0.001%, and other sectors 2.14%); industrial processes 22.95%; and waste 0.11% [41]. Taking into account the location of industrial plants, mainly located around cities, as well as the physicochemical properties of PCBs and the possibility of spreading these compounds in the environment, it can be assumed that the distance from the place of release of the compounds to the environment is of great importance in the assessment of human exposure. In this study, based on the redundancy analysis, positive relationships were found between the content of PCB 123 and PCB 167 and the place of residence. Other authors obtained similar dependencies [39,42,43]. Higher concentrations of PCBs in milk in urban areas may result from the continuous contamination of the environment with PCB compounds from landfills, transformers, and old landfills of PCB-containing equipment intended for disposal. However, PCBs are known for being long-distance transport contaminants as well as being bio-accumulative. Even in areas that are otherwise pristine locally (i.e., no industry), human populations have been found to have high levels of PCBs [44,45]. A lack of correlation between the place of residence and PCB concentration was also found by Dimitriadou et al. [33].

The relationship between PCB concentrations and the frequency of food consumption has also been reported by other researchers [46,47]. Parity can also influence PCB concentrations in human milk.

The study presented herein showed that there was a positive correlation between parity and the ndl-PCB content of human milk, which indicated that breastfeeding is one way of eliminating PCBs from the maternal body. Similar relationships have been reported by many researchers [35,48,49,50,51,52]. Hassine et al. [36] and Malarvannan et al. [53] reached different conclusions, as neither confirmed this relationship.

The average content of total ndl-PCBs in the milk tested in the present study from women residing in northwestern Poland was below the maximum level of 1.0 ng/g wet weight according to the relevant European Union regulations [24]. Given the changes in total ndl-PCBs throughout lactation (0.008–0.897 ng/g w.w.) in the current study, the maximum level of ndl-PCBs was not exceeded.

The range of dl-PCB content was 0.034–5.676 pg-TEQ/g w.w.; thus, assuming that the expression “food for infants and young children” [24] could refer to human milk, the maximum level for dl-PCBs was exceeded. When different lactation stages were considered, the maximum levels were not exceeded only in the milk collected after one year of breastfeeding. For example, the content of dl-PCBs in Sweden in 2011 was 0.068–0.086 pg TEQ/g w.w. [54], while in China in 2011 it was 0.014–0.158 pg TEQ/g w.w. [43]. In Italy in 2008–2009 the content of dl-PCBs in milk was 0.380 pg TEQ/g w.w. [55]. Slightly higher concentrations were reported in Belgium (0.024–0.330 pg TEQ/g w.w.) [56], in areas of Ireland (0.390 pg TEQ/g w.w.) [57], and in Hong Kong (0.410 pg TEQ/g w.w.) [58].

Based on data in the literature, male infants were estimated to consume an average of 76 mL more milk than female infants. Daily infant milk consumption fluctuated from 478 mL to 1356 mL at an average of 798 mL [25,26,27]. Considering the above, the weekly dl-PCB intake in human milk was within the range of 0.021–7.34 pg-TEQ/kg bw/week, which was an average of 2.12 pg-TEQ/kg bw/week for female infants and 2.18 pg-TEQ/kg bw/week for male infants.

The presence of PCBs in human milk, which might negatively influence child development, is evidence that exposure to them is not limited to the prenatal period (via placental transfer), but that it extends into the breastfeeding period, during which infants receive the largest pool of xenobiotics (dl-PCBs) in human milk in the first two months of life. There is potential for lowering the content of organochlorine compounds in breast milk by encouraging women expecting a child to adhere to certain nutritional recommendations, including avoiding the excessive consumption of fish from unknown origins and eliminating tobacco smoking.

## 4. Materials and Methods

### 4.1. Participants and Biological Study Material

Ninety-six mothers, aged 18–36 and residing in northwest Poland, participated in the study. None of the participants had had any previous contact with hazardous chemicals at their places of employment or residence. Women not included in the study were those for whom breastfeeding was contraindicated for reasons linked to either themselves or their infants, such as severe infant illness; serious illness that prevented the mother from caring for the infant; maternal cytotoxic chemotherapy; active maternal tuberculosis; excessive maternal alcohol consumption; etc.

Permission for the study was obtained from the Bioethics Commission at the Regional Chamber of Physicians and Dentists (Bioethics Commission permission no. OIL-Sz/MF/KB/452/02/04/2015, 23 April 2015). Based on our own experience, information from hospitals, and literature data, we developed our own 2-part questionnaire. The questionnaire collected information on the characteristics of the mothers participating in the study; the eating habits of the mothers during pregnancy, taking into account the frequency of the consumption of certain food products; and data on smoking (Table 1). The questionnaire also included information about the child. Before starting the research, mothers provided their written consent to participate in the experiment, i.e., to transfer samples of their own milk to the laboratory during particular periods of lactation and to fill in the questionnaire. The nutritional habits of the mothers participating in the study as reported on the questionnaire are presented in Figure 1. All participants provided the required information. None of the participants reported on the questionnaire that they had had previous contact with hazardous chemicals, inter alia, at their places of employment or residence.

Before the study began, each participant provided their written consent for the collection of their milk. All the mothers were supplied with glass bottles that had been previously rinsed with hexane and methanol used for chromatographic analyses. Human milk samples, in volumes of 40–100 mL, were poured from breast pumps directly into the prepared bottles. The bottles were closed with caps, which were additionally secured inside with aluminum foil. The milk samples were stored at −18 °C in glass bottles in freezers until they were shipped to the laboratory. The milk sampling schedule included the duration of lactation from the first day after birth to 12 months or more. A total of 920 samples were analyzed.

In the first month of lactation, samples were collected from the mothers weekly, i.e., on days 7, 14, 21, and 28 following childbirth. In subsequent months, the milk collected was a mean of four samples taken at the end of each month. Each of the mothers participating in the study received precise guidelines on how and when to collect milk. Additionally, each of the mothers reported the exact time of milk collection. More frequent milk collection in the first month of lactation compared to the remaining stages was prompted by the assumption that the greatest changes in PCB contents would occur during this time.

### 4.2. Chemical Materials—Reagents

All reagents used in the analyses (e.g., anhydrous sodium sulfate, hexane, and acetone) were high-purity-grade and were obtained from Merck (Germany).

The following standard solutions were used for the analyses:6 PCB-Seven Key Isomers NE 5575 (IUPAC No. PCBs 28, 52, 101, 118, 138, 153): LGC Ltd., Wesel, Germany.12 non-ortho and mono-ortho PCB-CERTAN© NE 5570 LGC Ltd. (IUPAC No. PCBs 77, 81, 123, 105, 114, 126, 156, 157, 180, 169, 167, 189): LGC Standards, Wesel, Germany.Reference material BCR 450-PCBs in natural milk powder: Community Bureau of Reference, Belgium.Pesticides Surrogate Spike Mix 4–8460, Supelco (Bellefonte, PA, USA).^13^C12-labeled PCB Mixture-A, CIL (Cambridge Isotope Laboratories, Inc.) EC-4938, containing: 3,3′,4,4′-TetraCB; 3,4,4′,5′-TetraCB; 2′,3,4,4′,5-PentaCB; 3,3′,4,4′,5-PentaCB; 3,3′,4,4′,5,5′-HexaCB; and 2,2′,3,4,4′,5,5′-HeptaCB.

### 4.3. Methodology

The study included:Estimating the contamination of milk with the following compounds: non-ortho PCBs (IUPAC No. 77, 81, 126, 169), mono-ortho PCBs (105, 114, 118, 156, 157), and PCB indicator congeners (28, 52, 101, 138, 153, 180).Assessing infant health risk with the parameters of TEQ_PCB_ and tolerable weekly intake (TWI).Analyzing the residues detected with regard to current regulations on maximum levels allowed in foods for infants and young children.

### 4.4. Analytical Methodology

The samples were homogenized and freeze-dried at −60 °C in a LYOLAB 3000. The freeze-dried samples were frozen at −18 °C and stored until analysis. Approximately 1.5 g of the freeze-dried material (in triplicate) was placed on extraction thimbles and extracted using a Soxhlet device (120 mL hexane-acetone, 3:1, *v*/*v*) for 8 h. Next, the samples were concentrated in a rotary vacuum evaporator to 2 mL, purified with 8 mL of fuming H_2_SO_4_ (7% SO_3_ in concentrated H_2_SO_4_ (*v*/*v*)), polished on a bed of basic alumina (1 g), and eluted with 14 mL 2% dichloromethane (DCM) in n-hexane and 17 mL 50% DCM in n-hexane. The samples were then concentrated under a stream of nitrogen to a volume of 0.5 mL. PCBs were determined with gas chromatography/mass spectrometry (HP 8890/5977 GC-MS) with the parameters described in Table 3.

The content of lipids was determined with the gravimetric method using 8 h extraction with an acetone/n-hexane mixture, (1/1, *v*/*v*), in a Soxhlet apparatus. Dry matter was estimated gravimetrically. The dry samples (dehydrated breast milk solids) were dried to a constant weight in an oven at a temperature of 105 °C. Protein and lactose were determined with a MilcoScan FT 6000 following the manufacturer’s instructions.

### 4.5. Quality Control

An internal standard solution (decachlorobiphenyl, 100 mL, 80 ng/mL; Pesticides Surrogate Spike Mix 4–8460, Supelco, USA) was used to determine recoveries. Additionally, PCB congener recoveries were estimated with isotope solution analysis (^13^C12-labeled PCB Mixture-A, CIL (Cambridge Isotope Laboratories, Inc.) EC-4938) containing the following: 3,3′,4,4′-TetraCB; 3,4,4′,5′-TetraCB; 2′,3,4,4′,5-PentaCB; 3,3′,4,4′,5-PentaCB; 3,3′,4,4′,5,5′-HexaCB; and 2,2′,3,4,4′,5,5′-HeptaCB (50 mL, 120 ng/mL). The mean recoveries of radiolabeled compounds ranged from 78.5% (PCB 81) to 102.1% (PCB 180). The recoveries of compounds for which radiolabeled labeled equivalents were unavailable were estimated by analyzing samples fortified with the compounds analyzed.

The following standard solutions were used for identification and quantification: 6 PCB-Seven Key Isomers LGC Ltd. NE 5575 (IUPAC No. PCB 28, 52, 101, 118, 138, 153); 12 PCB-CERTAN© NE 5570 LGC Ltd. (IUPAC No. PCB 77, 81, 123, 105, 114, 126, 156, 157, 180, 169, 167, 189). The recovery values of non-ortho and mono-ortho PCBs were, respectively, 77–77.8%, 126–79.6%, 169–87.4%, 114–78.0%; 156–79.0%, 157–82.9%, and 81–94.2%, while the range for ndl-PCBs was 77.2–88.2%. The recovery of decachlorobiphenyl (PCB 209), which was used as an internal standard, ranged from 84.1–98.3%.

To monitor the accuracy of the method, each batch of samples was also analyzed with reference material (reference material BCR 450-PCBs in natural milk powder, Community Bureau of Reference). The recovery of PCB congeners in the reference material (BCR 450-PCBs in natural milk powder) was determined against the mean declared value and was in the range of 87.9–98.2%. The limits of quantification (LOQ) of the PCB congeners were, on average, 0.1–0.15 ng/kg w.w.

The limit of detection (LOD) and LOQ were estimated based on the dependence of the standard deviation of the calibration curve intercept to its slope. The standard deviation of the calibration curve intercept at the LOD was multiplied by a value of 3.3, and at the LOQ by 10 [59,60]. The LOD and LOQ values were determined by analyzing matrix samples, blank samples, and matrix extracts with the addition of internal standards. The LOD for PCB congeners was estimated at 0.03–0.1 ng/kg w.w.

### 4.6. Estimated Health Risk

Infant exposure to PCBs present in human milk was estimated using toxic equivalents (TEQ) (Equation (1)).
TEQ_PCB_ = ∑_i_ (cPCB_i_ · TEF_i_),(1)
where:

TEQ—toxic equivalent 2,3,7,8-TCDD; cPCBi—concentrations of i-th non-ortho and mono-ortho PCB congeners; and TEFi—toxic equivalency factor of i-th PCB congeners (77—0.0001; 81—0.0003; 126—0.1; 169—0.03; 105—0.00003; 114—0.00003; 118—0.00003; 123—0.00003; 156—0.00003; 157—0.00003; 167—0.00003; and 189—0.00003) in reference to 2,3,7,8-TCDD [61].

### 4.7. Statistical Analysis

The study results were analyzed statistically with Statistica 10.0. Analysis of variance with Levene’s homogeneity test and the Kolmogorov–Smirnov normality test (K–S test) preceded the Anova test. Correlation coefficients and regression equations were calculated to determine correlations among the variables studied. The strength of correlations was interpreted with the Guilford classification [62]. Tukey’s test was used to assess the significance of differences (*p* < 0.05). The results are presented as arithmetic means, taking into account standard deviations (SD) and coefficients of variation (CV). Multivariate redundancy analysis was also used to process the results, which are presented in two-dimensional space on ordination diagrams.

## 5. Conclusions

The content of non-ortho, mono-ortho, and ndl-PCBs in human milk decreased throughout lactation. Neither maternal age nor parity significantly influenced the content of most of the xenobiotic compounds analyzed. The presence of both ndl-PCBs and dl-PCBs in human milk proved that exposure to them is not limited to the prenatal period (via placental transfer), but extends into the breastfeeding period. The present study indicated that infants receive the largest pool of the PCBs accumulated in milk in the first weeks of life. The study results also indicated that it is possible for pregnant women to lower the content of OCs by complying with specific nutritional recommendations, e.g., avoiding excessive fish consumption, especially those of unknown origin, and eliminating tobacco smoking.

Despite the content of organochlorine pollutants in human milk and the risks to infant health and further development, the World Health Organization emphasizes the enormous benefits of breastfeeding. However, the results of the current study underscore the necessity of continually monitoring organochlorine xenobiotics in human milk.

## Figures and Tables

**Figure 1 ijms-23-12717-f001:**
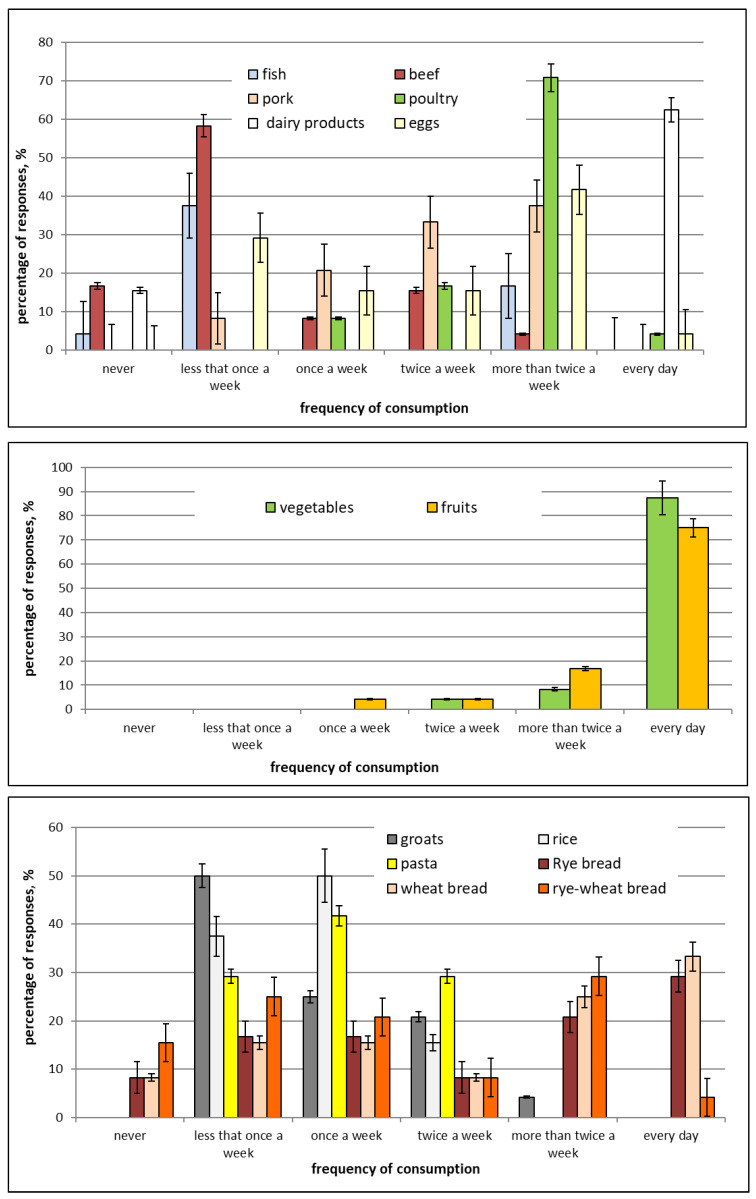
Eating habits of the pregnant women based on the survey.

**Figure 2 ijms-23-12717-f002:**
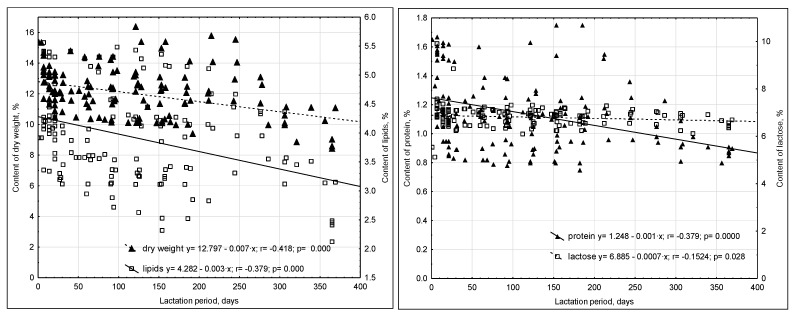
Changes in the contents of dry matter, lipids, protein, and lactose throughout lactation.

**Figure 3 ijms-23-12717-f003:**
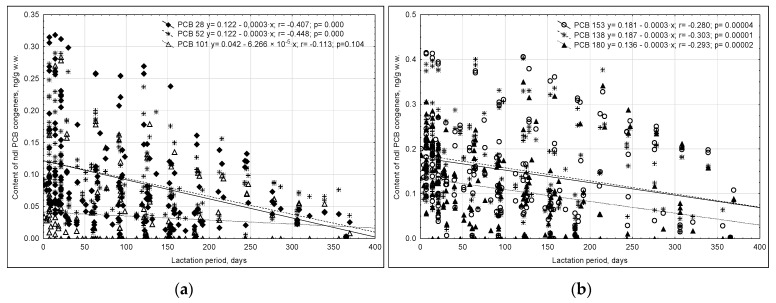
Changes in the content of ndl-PCB congeners in human milk throughout lactation: (**a**) PCB 28, PCB 52, PCB 101; (**b**) PCB 153, PCB 138, PCB 180.

**Figure 4 ijms-23-12717-f004:**
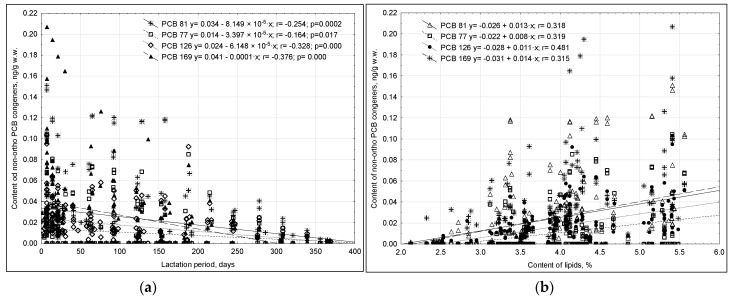
Correlation between the content of non-ortho PCBs in human milk and the duration of lactation (**a**) and milk lipid content (**b**).

**Figure 5 ijms-23-12717-f005:**
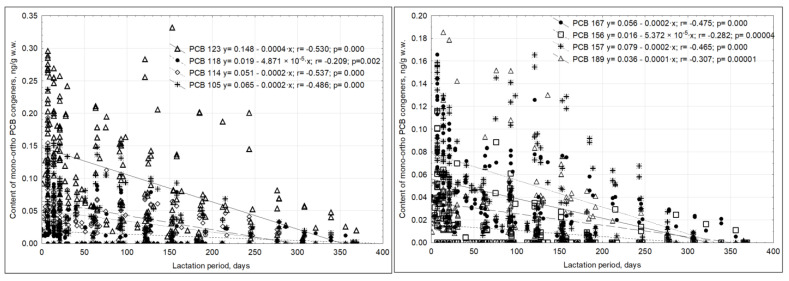
Correlations between the duration of lactation and the content of mono-ortho PCBs in human milk.

**Figure 6 ijms-23-12717-f006:**
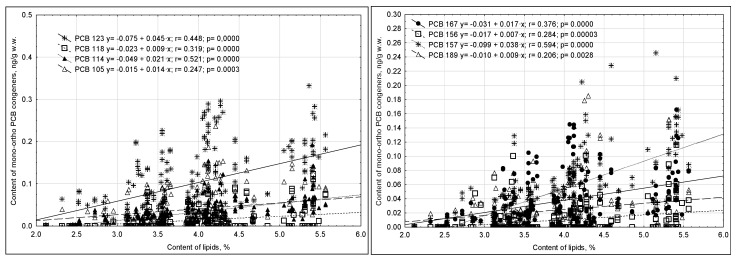
Correlations between mono-ortho PCB and lipid contents in the human milk tested.

**Figure 7 ijms-23-12717-f007:**
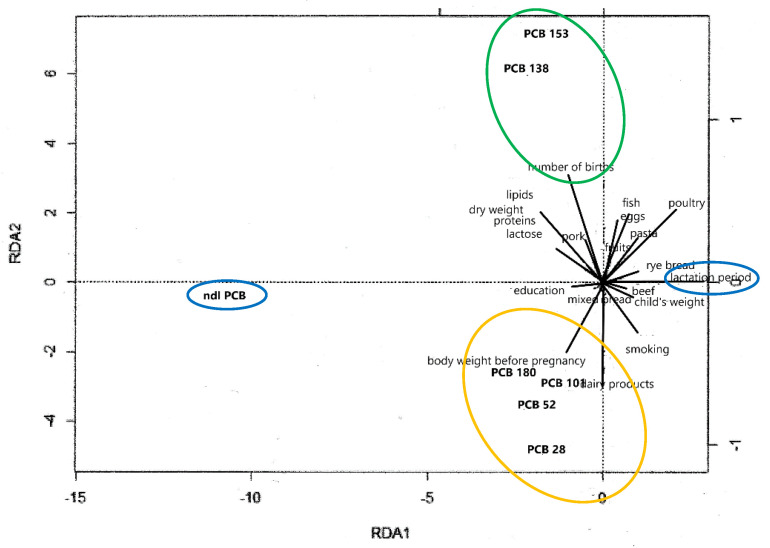
RDA biplot of relationships between ndl-PCBs and explanatory variables (the sets of objects are marked with colors, lines are the directions of the greatest variability).

**Figure 8 ijms-23-12717-f008:**
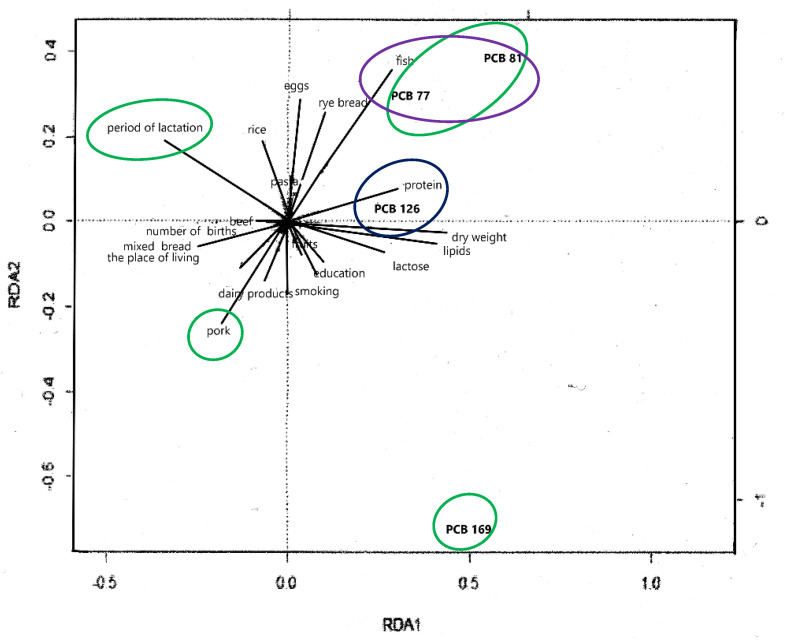
RDA biplot of relationships between non-ortho PCBs and explanatory variables (the sets of objects are marked with colors, lines are the directions of the greatest variability).

**Figure 9 ijms-23-12717-f009:**
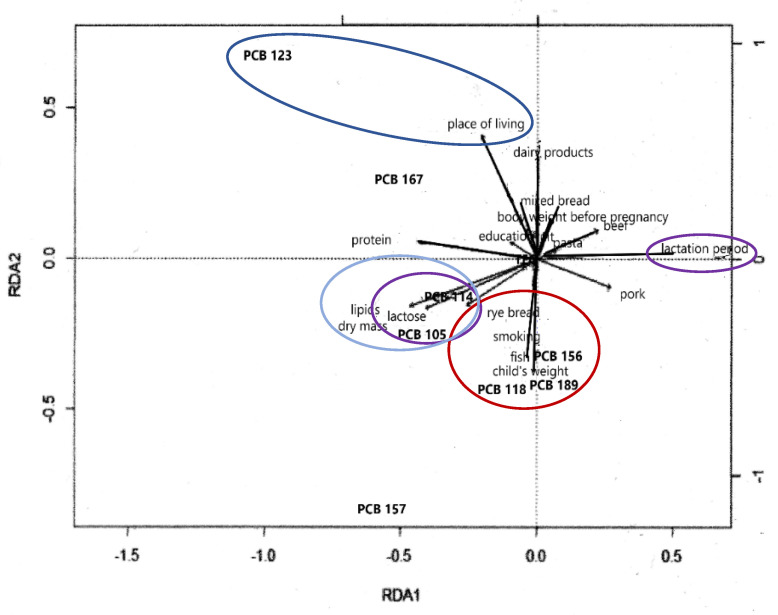
RDA biplot of relationships between mono-ortho PCBs and explanatory variables (the sets of objects are marked with colors, lines are the directions of the greatest variability).

**Figure 10 ijms-23-12717-f010:**
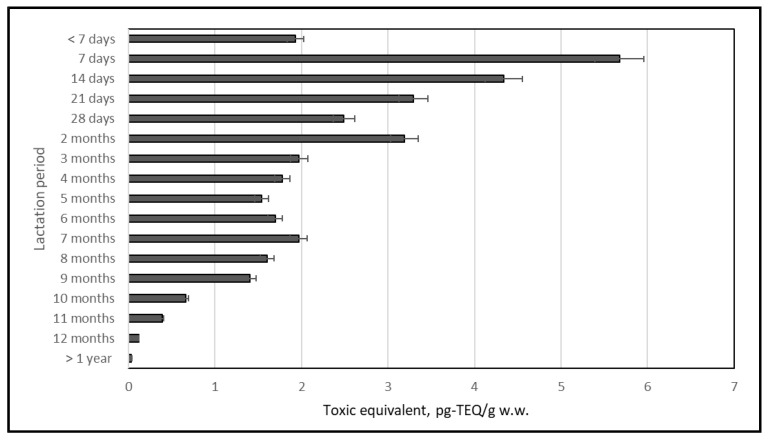
dl-PCBs in human milk.

**Figure 11 ijms-23-12717-f011:**
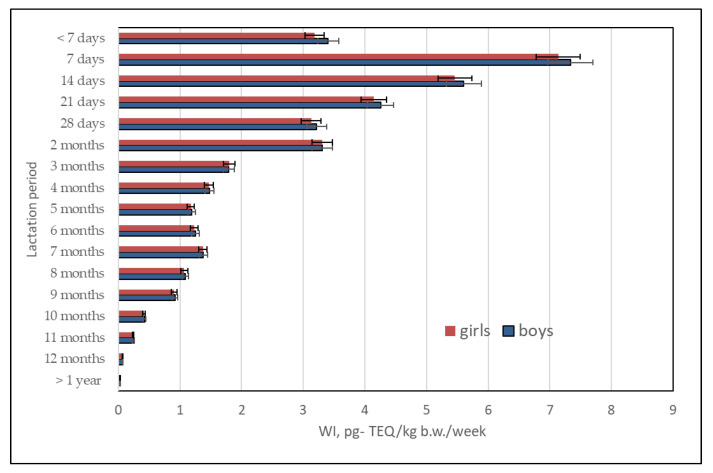
Estimated weekly intake of dl-PCBs in human milk.

**Table 1 ijms-23-12717-t001:** Characteristics of the pregnant women participating in the study (n^1^ = 96) [23].

No.	Basic Information on Mother and Child	Response Provided in Questionaire	Percentage of Provided Responses (%)
1.	mother’s age (years)	<18	4.2 (%)
		18–22	-
		23–27	25 (%)
		28–32	37.5 (%)
		33–36	33.3 (%)
2.	body weight prior to pregnancy (kg)	<55	25 (%)
		56–60	12.5 (%)
		61–65	20.8 (%)
		>65	41.7 (%)
3.	body weight in the first week after childbirth (kg)	55–60	20.8 (%)
		61–65	20.8 (%)
		66–70	25 (%)
		>70	33.4 (%)
4.	place of residence	city > 5000 inhabitants	83.3 (%)
		town < 5000 inhabitants	4.2 (%)
		country	12.5 (%)
5.	education	primary	4.2 (%)
		secondary	-
		higher	95.8 (%)
6.	number of childbirths performed	1	62.5 (%)
		2–3	37.5 (%)
		≥4	-
7.	type of birth	natural forces	50 (%)
		c-section	50 (%)
8.	infant’s sex	boy	62.5 (%)
		girl	37.5 (%)
9.	infant’s body weight in the first week (kg)	2.4–2.8	4.2 (%)
		2.9–3.2	45.8 (%)
		3.3–3.6	45.8 (%)
		3.7–4.0	-
		>4.0	4.2 (%)
10.	smoking	never smoked	75 (%)
		quitted before pregnancy	16.7 (%)
		used to smoke (number of packs of 20 cigarettes):	
		➢ up to one a day ➢ more than one a day	4.2 (%) 4.2 (%)

n^1^—number of mothers participating in the study.

**Table 2 ijms-23-12717-t002:** ndl-PCBs in human milk during lactation.

Sample Collection Period	Number of Samples	Σndl-PCB ng/g Lipids	CV ^b^	Σnon-*ortho* PCB ng/g Lipids	CV	Σmono-*ortho* PCB ng/g Lipids	CV
<7 days	17 ^a^	16.63 ± 1.37	8.24	1.71 ± 0.23	13.31	7.84 ± 0.75	9.52
7 days	26	19.83 ± 1.26	6.35	3.81 ± 0.50	13.15	14.85 ± 1.19	8.01
14 days	59	20.31 ± 1.36	6.70	3.04 ± 0.44	14.52	12.19 ± 1.04	8.53
21 days	98	17.94 ± 1.10	6.13	2.45 ± 0.32	13.00	9.85 ± 0.83	8.43
28 days	39	14.49 ± 0.74	5.11	2.37 ± 0.22	9.12	8.56 ± 0.94	10.98
2 months	75	15.90 ± 1.25	7.86	2.06 ± 0.20	9.81	8.99 ± 0.74	8.26
3 months	75	15.24 ± 1.14	7.48	1.65 ± 0.15	9.02	7.02 ± 0.61	8.65
4 months	92	15.22 ± 1.14	7.49	1.47 ± 0.18	12.22	6.21 ± 0.65	10.44
5 months	88	12.83 ± 0.97	7.56	1.39 ± 0.19	13.54	5.23 ± 0.55	10.43
6 months	88	10.82 ± 0.89	8.23	1.36 ± 0.09	6.72	3.37 ± 0.42	12.42
7 months	69	15.39 ± 1.46	9.49	1.25 ± 0.07	5.77	4.23 ± 0.38	8.95
8 months	39	15.71 ± 1.32	8.40	1.16 ± 0.10	8.89	3.55 ± 0.43	12.04
9 months	29	13.81 ± 1.28	9.27	1.15 ± 0.12	10.49	2.85 ± 0.37	12.84
10 months	29	11.63 ± 0.88	7.57	0.74 ± 0.12	16.02	2.35 ± 0.27	11.54
11 months	29	10.48 ± 0.72	6.87	0.42 ± 0.06	14.89	1.76 ± 0.22	12.75
12 months	29	7.75 ± 0.54	7.00	0.15 ± 0.03	18.82	1.02 ± 0.17	17.12
>1 year of feeding	39	0.343 ± 0.035	10.20	0.14 ± 0.05	32.16	0.21 ± 0.04	17.45
average content		13.78 ± 4.76		1.55 ± 0.98		5.89 ± 4.09	

^a^ Each sample was analyzed in triplicate; ^b^ CV—coefficient of variation.

**Table 3 ijms-23-12717-t003:** HP 8890/5977 GC–MS parameters.

Chromatograph Operating Parameters HP 8890/5977	
column furnace program	130 °C (0.5 min) → increase 7 °C/min → 200 °C (5 min) → 4 °C/min → 280 °C (10 min) → 290 °C (5 min)
injection in the column	2 µL
carrier gas	helium
flow through the column	1.1 mL/min
duration of the analysis	50 min

## Data Availability

Data available on request.

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
