# Peer review of "Changes in Polychlorinated Biphenyl Residues in Milk during Lactation: Levels of Contamination, Influencing Factors, and Infant Risk Assessment"

_ijms, 2022, doi:10.3390/ijms232112717_

Round 1

Reviewer 1 Report

Nutritional Habits of Women During Pregnancy, Polychlorinated Biphenyl Residues in Milk and Changes During Lactation

 The present study aimed to assess the quality of human milk in terms of polychlorinated biphenyl contaminations taking into account the duration of the lactation. The influence of other factors, such as the mother’s age and weight, place of residence, parity, and nutritional habits during pregnancy on the PCB levels in milk was analyzed.

Although the topic is of great interest, major changes are required.

In particular, a deeper discussion about some key points should be added to justify the obtained results and allow to use them in daily life.

In addition, also minor revisions are necessary.

Hereafter there’s a list of some questions and suggestions for your work.

Human Milk samples

From the presented data it is not clear how many milk samples of each lactation period are considered in the reported analyses.

In addition, no information on the analytical repetitions in the legends of the figures were reported.

Eating habits

The eating habits were investigated through a questionnaire but a lack of information regarding the methodology and the questionnaire used is evidenced. Authors have to clarify these aspects.

The acquired data are reported in figure 1, the graph is not well legible and there’s no x-axis title in the first part. In addition, the meaning of the bars must be explained.

Composition of human milk during lactation (section 2.2)

Data regarding the dry matter, lactose, lipids and protein content are not reported in the text.

In figure 2, the axis title “content of proteins, %” is missing. In addition, the dots used are too similar resulting in difficulties to catch the results.

Redundancy analysis (RDA) reports a relationship between PCBs content in milk and the quantities of consumed food (i.e. dairy products, fish, eggs) while the evaluation of the quantity of food is not described.

In addition, a positive correlation between the content of PCB 126 residues and the protein content of the milk is reported. How is it possible to discuss this result?  

Some studies cited in the discussion, revealed that the PCB levels in human milk is higher in women who consume local foodstuffs. Did the authors evaluate this aspect?

Furthermore, a negative correlation with women smoking cigarettes prior to pregnancy is reported, are the analyzed subjects enough to support this evidence? In addition, authors should underline if other factors (i.e. the duration of the exposure to smoke and/or the duration of the quitted period) can impact on PCBs presence in milk.

In figure 9, the RDA analysis confirmed a positive correlations between the concentrations of some PCB congeners and women’s residence in the cities with population greater than 5,000. Is it possible to discuss this evidence taking into account additional factors different from foods?

The maternal body weight prior pregnancy varied from less than 55kg to more than 65 Kg, could be useful to evaluate the BMI of the mothers instead of their body weight?

The data obtained in this study underline that the total daily intake of PCBs in infant could reach the TDI. In addition, authors published data regarding the presence and the quantification of other contaminants in human milk. Considering the presence of more than one residues and the resulting increased toxicity, seems important to add some considerations in the final discussion

On the basis of all the reported correlations between food consumption during pregnancy and milk contaminations, considering the nutrients need of a pregnant and of a woman during lactation is it possible to suggest something in terms of food choice to limit the PBC content in milk? These tips could be a useful milestone to further apply this knowledge on human health.

The authors have to read carefully the paper and correct some mistakes such as in line 53 children must be removed and in line 103 a space must be insert.

Last, but not least, the title is not in line with the aims of the study reported in the abstract

Author Response

Dear Reviewer,

all authors would like to thank you for your comments and time spent on the manuscript evaluation. Thank you for giving us the opportunity to revise and improve our article. All the comments and suggestions from the reviewers were extremely helpful and inspiring for us, thanks to which we could improve the quality of the manuscript. All changes have been saved in change tracking mode.

The authors have made the necessary corrections according to the reviewers suggestions.

Our answers to the reviewer:

From the presented data it is not clear how many milk samples of each lactation period are considered in the reported analyses

The data in table 2 have been supplemented

In addition, no information on the analytical repetitions in the legends of the figures were reported.

The data was supplemented in the manuscript

The eating habits were investigated through a questionnaire but a lack of information regarding the methodology and the questionnaire used is evidenced. Authors have to clarify these aspects.

Information in section 4.1 of the manuscript has been supplemented.

“Based on our own experience, information from hospitals and literature data, our own individual 2-part questionnaire was developed. The questionnaire contained the characteristics of the mothers participating in the study, the eating habits of mothers during pregnancy, taking into account the frequency of consumption of certain food products, and data on smoking. The questionnaire also included information about the child.. Before starting the research, mothers gave their written consent to participate in the experiment, i.e. to transfer samples of their own milk to the laboratory in particular periods of lactation and to fill in the questionnaire.”

The acquired data are reported in figure 1, the graph is not well legible and there’s no x-axis title in the first part. In addition, the meaning of the bars must be explained.

the figure 1 has been corrected to clearly present the presented data

Data regarding the dry matter, lactose, lipids and protein content are not reported in the text

requested information was supplemented in point 2.2

In figure 2, the axis title “content of proteins, %” is missing. In addition, the dots used are too similar resulting in difficulties to catch the results.

figure 2 has been corrected

Redundancy analysis (RDA) reports a relationship between PCBs content in milk and the quantities of consumed food (i.e. dairy products, fish, eggs) while the evaluation of the quantity of food is not described.

In this experiment, the amount of consumed food was not analyzed, but only the frequency of consumption of the most important groups of products. The study of the amount of eaten food was impossible because obtaining the consent of the respondents to participate in the research and filling in the questionnaire took place after the child was born, which did not allow for obtaining reliable information on the weight of the consumed products.

In addition, a positive correlation between the content of PCB 126 residues and the protein content of the milk is reported. How is it possible to discuss this result?  

No confirmation of the above dependence was found in the literature. Perhaps the reason can be found in the fact that PCB 126 has the structure most similar to dioxins, especially 2,3,7,8-TCDD, has the highest toxicity factor among PCBs and is characterized by the strongest affinity for binding to the Ah receptor, which is a protein aromatic hydrocarbon cytosolic receptor. This shows the affinity of dl PCBs for some types of proteins.

Some studies cited in the discussion, revealed that the PCB levels in human milk is higher in women who consume local foodstuffs. Did the authors evaluate this aspect?

The respondents of this study lived in the region of north-western Poland, which is located on the Baltic Sea. There are many fish processing plants in this area. It can therefore be argued that the consumed fish is a local product, the consumption of which may result in the increased content of PCBs in milk.

Furthermore, a negative correlation with women smoking cigarettes prior to pregnancy is reported, are the analyzed subjects enough to support this evidence? In addition, authors should underline if other factors (i.e. the duration of the exposure to smoke and/or the duration of the quitted period) can impact on PCBs presence in milk.

Required information was supplemented in point 2.4.

In figure 9, the RDA analysis confirmed a positive correlations between the concentrations of some PCB congeners and women’s residence in the cities with population greater than 5,000. Is it possible to discuss this evidence taking into account additional factors different from foods?

The figures 7, 8 and 9 have been corrected.

The main source of exposure of the general population (apart from occupational exposure) to PCBs is food (90-95%). The presence of these compounds in food, including human milk, results from the infiltration of these pollutants from the environment. The share of dominant sectors in PCB emissions in 2020 in Poland was as follows: fuel combustion 76.94% (energy industry 69.2%, manufacturing industry and construction 5.6%, transport 0.001% and other sectors 2.14%), industrial processes 22.95% and waste 0.11% [National emission balance of SO2, NOX, CO, NH3, NMVOC, dust, heavy metals and POPs for the years 1990 - 2020. Synthetic report. National Center for Emission Management and Balancing, Institute of Environmental Protection - National Research Institute. Ministry of Climate and Environment, Warsaw 2022]. Taking into account the location of industrial plants, mainly located around cities, as well as the physicochemical properties of PCBs and the possibility of spreading these compounds in the environment, it can be assumed that the distance from the place of release of compounds to the environment is of great importance in the assessment of human exposure.

The maternal body weight prior pregnancy varied from less than 55 kg to more than 65 kg, could be useful to evaluate the BMI of the mothers instead of their body weight?

The aim of the survey was not to assess the overweight or obesity of future mothers, therefore the BMI index was omitted.

The data obtained in this study underline that the total daily intake of PCBs in infant could reach the TDI. In addition, authors published data regarding the presence and the quantification of other contaminants in human milk. Considering the presence of more than one residues and the resulting increased toxicity, seems important to add some considerations in the final discussion.

The discussion was supplemented and the text of the work was amended.

On the basis of all the reported correlations between food consumption during pregnancy and milk contaminations, considering the nutrients need of a pregnant and of a woman during lactation is it possible to suggest something in terms of food choice to limit the PBC content in milk? These tips could be a useful milestone to further apply this knowledge on human health.

There is a potential possibility of lowering the content of organochlorine compounds in breast milk by adhering to certain nutritional recommendations by women expecting a child, including avoiding excessive consumption of fish, especially of unknown origin, and eliminating smoking.

The authors have to read carefully the paper and correct some mistakes such as in line 53 children must be removed and in line 103 a space must be insert.

The indicated errors have been corrected

Kind regards,

Kamila Pokorska-Niewiada,

corresponding author

Reviewer 2 Report

This is the revision of the manuscript with the title " Nutritional Habits of Women During Pregnancy, Polychlorinated Biphenyl Residues in Milk and Changes During Lactation" by Witczak & colleagues. In general, the concept of the manuscript is interesting. The text is well written and well organised. The work is appropriate for the Journal and the results are interesting. The elaborated methods were fully validated and optimized in terms of analytical chemistry approach. However, there are some observations that must be addressed before the manuscript can be accepted.

A serious downside is the lack of emphasis on scientific novelty. There are many reports in the literature on the PCBs content in human milk and the variability of their concentrations over time. It is necessary to emphasize the novelty of the results gained.

Figure 1. first graph – change the a axis in the present form, it is difficult to recognize which bars correspond to twice a week, more than twice a week and every day. Graph 2 – error bar is cut off

Figure 2, 3,4, 5,6. I recommend changing the points on the charts. In present form, it is difficult to differ either it is circle or a square.

Figure 7, 8,9, 11. Quality needs to be improved. It is difficult to read in its present form.

In Table 2 and through the text, write the results down to two significant digits.

L364, 365: write the formulas using subscript

L372: it would be useful to provide a methodology for lipid content and dry matter determination in supplementary materials.

L404: change the LOD unit to be in accordance with others used in the manuscript.

I would expect more in terms of discussing these results in the light of other results and conclusions. A deeper discussion is needed. Moreover, presentation of the selected chromatogram obtained would increase the scientific value of the presented study

Moreover, all references should be checked for appropriateness in the context in which they are cited and compliance with the required style.

Author Response

Dear Reviewer,

all authors would like to thank you for your comments and time spent on the manuscript evaluation. Thank you for giving us the opportunity to revise and improve our article. All the comments and suggestions from the reviewers were extremely helpful and inspiring for us, thanks to which we could improve the quality of the manuscript. All changes have been saved in change tracking mode.

The authors have made the necessary corrections according to the reviewers suggestions.

Our answers to the reviewer 2:

A serious downside is the lack of emphasis on scientific novelty. There are many reports in the literature on the PCBs content in human milk and the variability of their concentrations over time. It is necessary to emphasize the novelty of the results gained.

Human milk provides vital protection and development of the baby by giving all the essential components, such as macronutrients, vitamins, minerals, long-chain polyunsaturated fatty acids and cytokines.  Human milk is recommended as the sole source of nutrition for all babies during the first six months of life. Unfortunately, despite these benefits, it also tends to accumulate lipophilic toxins, especially persistent organic pollutants including polychlorinated biphenyls. Their ubiquitous distribution in various elements of environment are affected directly on the endocrine and reproductive systems in human. PCB can reduce fertility, interfere with hormone secretion or activity.

The assessing of the infants exposure to residues of ndl PCB and dl PCB in human milk is the first study of this type in north-western Poland. In this study assumed also the concentration of selected Polychlorinated biphenyls in human breast milk may have association with the individual features of the mother and her infant, and especially dietary habits.

Figure 1. first graph – change the a axis in the present form, it is difficult to recognize which bars correspond to twice a week, more than twice a week and every day. Graph 2 – error bar is cut off.

figures 1 and 2 have been corrected

Figure 2, 3, 4, 5, 6. I recommend changing the points on the charts. In present form, it is difficult to differ either it is circle or a square.

figures 2, 3, 4, 5, 6 have been corrected

Figure 7, 8,9, 11. Quality needs to be improved. It is difficult to read in its present form.

figures 7, 8, 9, 11 have been corrected

In Table 2 and through the text, write the results down to two significant digits

The data in table 2 have been supplemented and corrected

L364, 365: write the formulas using subscript.

Corrections in chemical formulas were made

L372: it would be useful to provide a methodology for lipid content and dry matter determination in supplementary materials

Requested information was supplemented in point 4.4 of the work.

L404: change the LOD unit to be in accordance with others used in the manuscript.

Requested information was corrected in manuscript

I would expect more in terms of discussing these results in the light of other results and conclusions. A deeper discussion is needed. Moreover, presentation of the selected chromatogram obtained would increase the scientific value of the presented study.

The chapters results, discussion and conclusions were supplemented

Moreover, all references should be checked for appropriateness in the context in which they are cited and compliance with the required style.

All references were checked for appropriateness in the context in which they are cited and compliance with the required style.

Kind regards,

Kamila Pokorska-Niewiada,

corresponding author

Reviewer 3 Report

This article is a welcome addition to the toxicological literature investigating the burdens of legacy contaminants in humans, in this case, PCBs in breast milk in Poland. The study is well executed, and the results show interesting relationships between PCB levels and breastfeeding over time, as well as other influences such as diet. It includes a risk assessment of the levels of PCBs in breast milk, based on food guidelines for infants and young children.

General comments: While excellent, all the graphs are extremely busy with all the individual points represented, which makes them difficult to read. Is there a way to separate the congeners into different graphs so that the data is more legible? 

Line 13-14 - Suggest "... to assess the contamination of human milk with dioxin-like PCBs..."

Line 39-40 - Suggest "PCBs are also known to be endocrine disruptors..."

Line 41-42 - This sentence needs to be rephrased. Is the author saying that triiodothyronine and thyroxine molecules have similar structure to certain PCBs, therefore leading to potential thyroid hormone disruptions?

Line 41-45 - Much information is provided in these lines without references - are all these statements associated with reference 4 or 5?

Line 57 - Suggest "...sperm that worsens over time..." if that is the accurate interpretation.

Line 63-65 - This sentence also could be edited, as it is unclear what the author is trying to convey. The amount of PCBs in a mother's body, and therefore blood, is directly related to the amount of PCBs a fetus will be exposed to in utero. However, the discussion of the involvement of the placenta is not clear.

Line 70-73 - This sentence needs a reference citation.

Line 77-78 - There may be cultural differences in the availability of, access to, and perceived invasiveness of collecting breast milk for PCB biomonitoring worldwide. Also, doing PCB biomonitoring in breast milk omits large portions of the population. 

Line 85-86 - While true, this leads the reader to assume that the author is proposing guidelines be established for permissible concentrations of PCBs (and presumably other deleterious substances) in human breast milk. How realistic or enforceable is that? How desirable or necessary is it? Might it not be better to virtually eliminate all sources of PCBs?

Line 91-95 - This paragraph could use some clarification of the language.

Table 1 - For "type of birth" category, suggest "Vaginal" rather than "Natural forces"

Line 134 - It is unclear what is being referred to in the term "dry matter". Could the author clarify what is contained in the dry matter? Dehydrated breast milk solids? Inorganic matter?

Line 158 - Typo - missing a % in the "7 days of feeding the infant" number report.

Table 2 - Header typos for non-ortho and mono-ortho PCB.

Line 288 - Typo - Figure 4, not 3.

Line 317-323 - Check the formatting.

Line 320 - What is wet matter? Non-lipid solids from the breast milk? How is that separated?

Line 346 - Typo, missing word "...81 in milk is strongly positively correlated..."

Line 372-373 - What food is this guideline set for? Baby formula? This would be the most analogous to breast milk that would be appropriate for comparison. Is it appropriate to use food guidelines for breast milk?

Line 377 - Typo of an orphan opening quotation mark (") mid-sentence

Figure 11 - Would it be possible to show the data as dietary intake per week so it would be directly comparable with the TDI? Then the TDI guideline can be overlaid over the data for easy visualization.

Line 421-422 - Without having the origin of fish reported within the study, this sentence is conjecture. Better to reframe along the lines of fish from the Baltic Sea are known to have high levels of PCB contamination, and therefore if the women in the study were eating local fish, this could explain the relationship between fish consumption and higher levels of PCBs.

Line 430-434 - The location argument is not strong, as PCBs are known as long distance transport contaminants as well as being bioaccumulative. Even in geographical places that are otherwise pristine locally (i.e., no industry), human populations have been measured with high levels of PCBs (Long et al 2021 (https://pubmed.ncbi.nlm.nih.gov/33803338/); Bjerregard et al 2013 (https://pubmed.ncbi.nlm.nih.gov/23562682/)).

Line 439-440 - This sentence is unclear and a double negative - did Dimitriadou et al indeed find a correlation between place and PCB concentration?

Line 443 - Suggest "The study presented herein showed there is a positive..." Prove is not a good verb for science.

Line 450 - Suggest "chosen" rather than "applicable" as the EU regulation does not apply to breast milk, rather it applies to food for infants and young children.

Line 453-454 - Suggest reframing, as the authors are using the EU regulation as an example, and the regulation cannot be assumed to apply to human breast milk.

Line 453-461 - The list of other countries' reported PCB concentrations should be linked to the concentrations reported in this study at the beginning of this paragraph. The reviewer infers from the list that the authors wish to show that Poland has the highest levels of the countries listed. This should be stated. Also, the author could investigate the levels reported in the USA, Greenland, Canada, Russia, and elsewhere as well.

Line 474-477 - A strong relationship was also shown for certain congeners with dairy and maternal weight, and negatively correlated with fish consumption. Also it is stated that PCBs 153 and 138 were negatively correlated with women smoking cigarettes. While the reviewer agrees that smoking should not occur during pregnancy for many well known reasons, including their toxic contaminant content, the reviewer would caution the authors on making medical recommendations, such as avoiding fish during pregnancy, based on the current data.

Line 480-486 - When were the mothers enrolled in the study? During the first trimester? At the birth of the child?

Line 487 - The "A total of 920 samples were analyzed." is out of place at the beginning of this paragraph. Reorganize for logic.

Line 491 - Suggest "The questionnaire collected information on the characteristics of..."

Line 491-497 - Was the questionnaire completed throughout the pregnancy or after birth (i.e., was the food diary/diet survey done as a recall survey)?

Line 501-502 - Repeated sentences.

Line 575-577 - It is still unclear what the dry matter consists of - solely dehydrated breast milk? All portions of the breast milk aside from water and volatiles that would have been driven off by 105'C? 

Line 577 - Repetition of "to constant weight"

Line 589-592 - Need to clarify the recoveries reporting. What does each range refer to?

Author Response

Dear Reviewer,

all authors would like to thank you for your comments and time spent on the manuscript evaluation. Thank you for giving us the opportunity to revise and improve our article. All the comments and suggestions were extremely helpful and inspiring for us, thanks to which we could improve the quality of the manuscript. All changes have been saved in change tracking mode.

Our answers:

Line 13-14 - Suggest "... to assess the contamination of human milk with dioxin-like PCBs..."

-corrected as recommended by the reviewer

Line 39-40 - Suggest "PCBs are also known to be endocrine disruptors..."

-corrected as recommended by the reviewer

Line 41-42 - This sentence needs to be rephrased. Is the author saying that triiodothyronine and thyroxine molecules have similar structure to certain PCBs, therefore leading to potential thyroid hormone disruptions?

-corrected as recommended by the reviewer

Line 41-45 - Much information is provided in these lines without references - are all these statements associated with reference 4 or 5?

- corrected

Line 57 - Suggest "...sperm that worsens over time..." if that is the accurate interpretation.

- corrected

Line 63-65 - This sentence also could be edited, as it is unclear what the author is trying to convey. The amount of PCBs in a mother's body, and therefore blood, is directly related to the amount of PCBs a fetus will be exposed to in utero. However, the discussion of the involvement of the placenta is not clear.

- corrected

Line 70-73 - This sentence needs a reference citation.

- citation added 

Line 77-78 - There may be cultural differences in the availability of, access to, and perceived invasiveness of collecting breast milk for PCB biomonitoring worldwide. Also, doing PCB biomonitoring in breast milk omits large portions of the population. 

-supplemented in the manuscript text

Line 85-86 - While true, this leads the reader to assume that the author is proposing guidelines be established for permissible concentrations of PCBs (and presumably other deleterious substances) in human breast milk. How realistic or enforceable is that? How desirable or necessary is it? Might it not be better to virtually eliminate all sources of PCBs?

-The lack of MRL guidelines for PCBs in human milk is currently a problem as it is the most important food in the first stage of life, when the body is most sensitive to contamination. Unfortunately, the rapid total reduction of PCBs in the environment, and hence in food, does not occur as quickly as would be expected. In addition, the change of the carcinogenicity category to 1 and the reduction of TDI values indicate the need to determine the MRL in human milk.

Line 91-95 - This paragraph could use some clarification of the language.

-corrected as recommended by the reviewer

Table 1 - For "type of birth" category, suggest "Vaginal" rather than "Natural forces"

– after consulting a native speaker, we will stick to the current nomenclature.

Line 134 - It is unclear what is being referred to in the term "dry matter". Could the author clarify what is contained in the dry matter? Dehydrated breast milk solids? Inorganic matter? - :dry matter” means “dehydrated breast milk solids”

Line 158 - Typo - missing a % in the "7 days of feeding the infant" number report.

-corrected as recommended by the reviewer

Table 2 - Header typos for non-ortho and mono-ortho PCB.

-corrected

Line 288 - Typo - Figure 4, not 3.

-corrected

Line 317-323 - Check the formatting.

-corrected

Line 320 - What is wet matter? Non-lipid solids from the breast milk? How is that separated?

-The term "wet matter" means raw milk, homogenized without separation

Line 346 - Typo, missing word "...81 in milk is strongly positively correlated..."

-corrected

Line 372-373 - What food is this guideline set for? Baby formula? This would be the most analogous to breast milk that would be appropriate for comparison. Is it appropriate to use food guidelines for breast milk?

-There are no MRLs defined for baby formulas In Poland, but only for "maximum levels permitted in foods for infants and young children" - according to Commission regulation (EU) No 1259/2011.

Line 377 - Typo of an orphan opening quotation mark (") mid-sentence

-corrected

Figure 11 - Would it be possible to show the data as dietary intake per week so it would be directly comparable with the TDI? Then the TDI guideline can be overlaid over the data for easy visualization.

-the table 11 was changed in line with the reviewer's recommendations

Line 421-422 - Without having the origin of fish reported within the study, this sentence is conjecture. Better to reframe along the lines of fish from the Baltic Sea are known to have high levels of PCB contamination, and therefore if the women in the study were eating local fish, this could explain the relationship between fish consumption and higher levels of PCBs.

-Thank you for your suggestion, such research will be the subject of future research / publication

Line 430-434 - The location argument is not strong, as PCBs are known as long distance transport contaminants as well as being bioaccumulative. Even in geographical places that are otherwise pristine locally (i.e., no industry), human populations have been measured with high levels of PCBs (Long et al 2021 (https://pubmed.ncbi.nlm.nih.gov/33803338/); Bjerregard et al 2013 (https://pubmed.ncbi.nlm.nih.gov/23562682/)).

- the text was supplemented with the indicated references

Line 439-440 - This sentence is unclear and a double negative - did Dimitriadou et al indeed find a correlation between place and PCB concentration?

- corrected sentence

Line 443 - Suggest "The study presented herein showed there is a positive..." Prove is not a good verb for science.

 -corrected

Line 450 - Suggest "chosen" rather than "applicable" as the EU regulation does not apply to breast milk, rather it applies to food for infants and young children.

-corrected

Line 453-454 - Suggest reframing, as the authors are using the EU regulation as an example, and the regulation cannot be assumed to apply to human breast milk.

-corrected

Line 453-461 - The list of other countries' reported PCB concentrations should be linked to the concentrations reported in this study at the beginning of this paragraph. The reviewer infers from the list that the authors wish to show that Poland has the highest levels of the countries listed. This should be stated. Also, the author could investigate the levels reported in the USA, Greenland, Canada, Russia, and elsewhere as well.

 – It was not our goal to show the highest content in Poland, only examples from other countries have been indicated.

Line 474-477 - A strong relationship was also shown for certain congeners with dairy and maternal weight, and negatively correlated with fish consumption. Also it is stated that PCBs 153 and 138 were negatively correlated with women smoking cigarettes. While the reviewer agrees that smoking should not occur during pregnancy for many well known reasons, including their toxic contaminant content, the reviewer would caution the authors on making medical recommendations, such as avoiding fish during pregnancy, based on the current data.

-corrected

Line 480-486 - When were the mothers enrolled in the study? During the first trimester? At the birth of the child?

-Due to the difficulty in obtaining the studied material (during the entire lactation period), the search for women who could and would like to take part in the research was carried out among future mothers, i.e. pregnant women, as well as those who had just given birth.

Line 487 - The "A total of 920 samples were analyzed." is out of place at the beginning of this paragraph. Reorganize for logic.

-corrected

Line 491 - Suggest "The questionnaire collected information on the characteristics of..."

-corrected

Line 491-497 - Was the questionnaire completed throughout the pregnancy or after birth (i.e., was the food diary/diet survey done as a recall survey)?

-The questionnaire was completed throughout pregnancy or after delivery, depending on the date on which the participant's consent to participate in the study was signed.

Line 501-502 - Repeated sentences.

-corrected

Line 575-577 - It is still unclear what the dry matter consists of - solely dehydrated breast milk? All portions of the breast milk aside from water and volatiles that would have been driven off by 105oC? 

-yes

Line 577 - Repetition of "to constant weight"

-corrected

Line 589-592 - Need to clarify the recoveries reporting. What does each range refer to?

-corrected

Kind regards,

Kamila Pokorska-Niewiada

Round 2

Reviewer 1 Report

The article was extensively revised from the authors on the light of the reviewer’s comments.

However, some problems remain:

With regards to food habits investigation a methodological limitation is evident.

Authors declare to utilize a questionnaire developed from their own, and not a validate tool. In addition, mothers filling the questionnaire reporting their food habits during pregnancy after the child born, the acquire data are not reliable and not adequate to sustain a paper titled “Nutritional Habits of Women During Pregnancy, Polychlorin- 2 ated Biphenyl Residues in Milk and Changes During Lactation”

Furthermore, Figure 1 is not in line with the reported results.

Regarding the comment:

Redundancy analysis (RDA) reports a relationship between PCBs content in milk and the quantities of consumed food (i.e. dairy products, fish, eggs) while the evaluation of the quantity of food is not described.

Authors report  “In this experiment, the amount of consumed food was not analyzed, but only the frequency of consumption of the most important groups of products. The study of the amount of eaten food was impossible because obtaining the consent of the respondents to participate in the research and filling in the questionnaire took place after the child was born, which did not allow for obtaining reliable information on the weight of the consumed products.”

but in the text the “quantities of food” still remain (i.e. line 334 and 342).

Regarding the comment:

“On the basis of all the reported correlations between food consumption during pregnancy and milk contaminations, considering the nutrients need of a pregnant and of a woman during lactation is it possible to suggest something in terms of food choice to limit the PBC content in milk? These tips could be a useful milestone to further apply this knowledge on human health”

Authors report:

There is a potential possibility of lowering the content of organochlorine compounds in breast milk by adhering to certain nutritional recommendations by women expecting a child, including avoiding excessive consumption of fish, especially of unknown origin, and eliminating smoking.

The nutritional recommendations are not fully developed.

Author Response

Dear Reviewer,

all authors would like to thank you for your comments and time spent on the manuscript evaluation. Thank you for giving us the opportunity to revise and improve our article. All the comments and suggestions were extremely helpful and inspiring for us, thanks to which we could improve the quality of the manuscript. All changes have been saved in change tracking mode.

Our answers:

REVIEWER 1 (ROUND 2)

Authors declare to utilize a questionnaire developed from their own, and not a validate tool. In addition, mothers filling the questionnaire reporting their food habits during pregnancy after the child born, the acquire data are not reliable and not adequate to sustain a paper titled “Nutritional Habits of Women During Pregnancy, Polychlorin- 2 ated Biphenyl Residues in Milk and Changes During Lactation”

- the authors modified the title of the work

Furthermore, Figure 1 is not in line with the reported results.

- corrected

Regarding the comment: Redundancy analysis (RDA) reports a relationship between PCBs content in milk and the quantities of consumed food (i.e. dairy products, fish, eggs) while the evaluation of the quantity of food is not described. Authors report  “In this experiment, the amount of consumed food was not analyzed, but only the frequency of consumption of the most important groups of products. The study of the amount of eaten food was impossible because obtaining the consent of the respondents to participate in the research and filling in the questionnaire took place after the child was born, which did not allow for obtaining reliable information on the weight of the consumed products.”

-the authors transformed the content of the work in accordance with the reviewer's recommendations

in the text the “quantities of food” still remain (i.e. line 334 and 342).

- corrected

 “On the basis of all the reported correlations between food consumption during pregnancy and milk contaminations, considering the nutrients need of a pregnant and of a woman during lactation is it possible to suggest something in terms of food choice to limit the PBC content in milk? These tips could be a useful milestone to further apply this knowledge on human health” Authors report: There is a potential possibility of lowering the content of organochlorine compounds in breast milk by adhering to certain nutritional recommendations by women expecting a child, including avoiding excessive consumption of fish, especially of unknown origin, and eliminating smoking.The nutritional recommendations are not fully developed.

-the authors supplemented the manuscript

Kind regards,

Kamila Pokorska-Niewiada
